# The CWI Pathway: A Versatile Toolbox to Arrest Cell-Cycle Progression

**DOI:** 10.3390/jof7121041

**Published:** 2021-12-04

**Authors:** Inma Quilis, Mercè Gomar-Alba, Juan Carlos Igual

**Affiliations:** Departament de Bioquímica i Biologia Molecular, Institut Universitari de Biotecnologia i Biomedicina (BIOTECMED), Universitat de València, 46100 València, Spain; inmaculada.quilis@uv.es (I.Q.); merce.gomar@uv.es (M.G.-A.)

**Keywords:** cell cycle, cell wall integrity, Pkc1, Slt2, checkpoint, DNA damage

## Abstract

Cell-signaling pathways are essential for cells to respond and adapt to changes in their environmental conditions. The cell-wall integrity (CWI) pathway of *Saccharomyces cerevisiae* is activated by environmental stresses, compounds, and morphogenetic processes that compromise the cell wall, orchestrating the appropriate cellular response to cope with these adverse conditions. During cell-cycle progression, the CWI pathway is activated in periods of polarized growth, such as budding or cytokinesis, regulating cell-wall biosynthesis and the actin cytoskeleton. Importantly, accumulated evidence has indicated a reciprocal regulation of the cell-cycle regulatory system by the CWI pathway. In this paper, we describe how the CWI pathway regulates the main cell-cycle transitions in response to cell-surface perturbance to delay cell-cycle progression. In particular, it affects the Start transcriptional program and the initiation of DNA replication at the G1/S transition, and entry and progression through mitosis. We also describe the involvement of the CWI pathway in the response to genotoxic stress and its connection with the DNA integrity checkpoint, the mechanism that ensures the correct transmission of genetic material and cell survival. Thus, the CWI pathway emerges as a master brake that stops cell-cycle progression when cells are coping with distinct unfavorable conditions.

Cell proliferation is the basis of life propagation. For this reason, the cell cycle is a crucial process subject to sophisticated molecular control. At the heart of this regulatory system is a family of kinases called CDKs (cyclin-dependent kinases), whose activity oscillates throughout the cell cycle depending on their association with activator proteins (cyclins), inhibitory proteins (CDK inhibitor proteins or CKIs), and the phosphorylation status of certain Thr and Tyr residues. The existing levels of cyclins and CKIs, as well as many other proteins involved in cell-cycle processes, oscillate along the cell cycle. Two main molecular mechanisms are responsible for this: regulation of gene transcription, with the existence of successive waves of gene expression along the different cell-cycle stages, and cell-cycle regulation of protein degradation by the ubiquitin–proteosome pathway [1].

Major control points in cell-cycle regulation occur at the G1/S transition (a key control point in which cells commit to a new round of cell division, initiating DNA replication), the G2/M transition (or entry into mitosis), and the metaphase–anaphase transition, coupled with the exit of mitosis. Errors in cell-cycle processes or cell-cycle progression under inadequate conditions can have catastrophic consequences for the cells. For this reason, eukaryotic cells have developed complex surveillance mechanisms, referred to as checkpoints, to ensure faithful cell division [2]. Checkpoints monitor that internal and external conditions are favorable. Once a deleterious condition is detected, checkpoints transiently arrest cell-cycle progression until the situation is amended. The components of the checkpoint mechanism are revealed by mutations that fail to stop the cell cycle, putting cell viability at risk.

Several signaling pathways transduce external and internal signals to the cell machinery to adapt the cellular physiology. One of the main signaling pathways in *Saccharomyces cerevisiae* is the cell-wall integrity (CWI) pathway. The major functions of the pathway are control of cell-wall biosynthesis and actin cytoskeleton dynamics, responding to environmental stresses and morphogenetic processes that affect the cell surface. Briefly, Pkc1 is activated by GTPase Rho1 after a perturbation in the cell wall is detected by membrane-sensor proteins. Pkc1, in turn, phosphorylates and activates a mitogen-activated protein kinase (MAPK) module comprising the MAPKKK Bck1, the redundant pair MAPKK Mkk1 and Mkk2, and the MAPK Slt2/Mpk1, which ultimately activates the transcriptional factor Rlm1 in order to induce the expression of genes involved in cell-wall biogenesis [3].

## 1. *PKC1*: A Cell-Cycle Gene?

The relationship between the CWI pathway and the cell cycle has been a remarkable aspect since the beginning of research into this pathway. Protein kinase C (Pkc1) was the first discovered component of the CWI signaling pathway, presented as an essential protein for the yeast cell cycle [4]. As is characteristic for cell division cycle (*cdc*) mutants, *pkc1* mutant cells arrested with a uniform terminal phenotype, which indicates a defect in a stage-specific function of the cell cycle. In particular, *pkc1* mutant cells accumulate in G2 as small budded cells, with a single nucleus and replicated DNA. At that time, it was suggested that *PKC1* might control an unknown checkpoint in the cell cycle. Immediately, subsequent studies reported that the lethality of *pkc1* mutant cells was due to important cell-wall defects, causing cell lysis during budding [5,6]. This raised the question of whether Pkc1 was really a relevant gene for cell-cycle control, or whether its terminal phenotype simply reflected that its defect in cell-wall biosynthesis was critical at a specific point in the cell cycle. During the following years, however, accumulated evidence has supported that the connection of Pkc1 and the CWI pathway with the cell cycle goes beyond the expected cell-cycle regulation of the pathway, given that cell-wall biosynthesis and actin cytoskeleton dynamics are cell-cycle-regulated processes. In fact, the CWI pathway acts as a bridge to interweave the morphogenetic processes that it regulates with the cell-cycle control machinery. Furthermore, it also acts on other aspects of the cell cycle, which are apparently unrelated to the control of the cell wall.

The characteristics and functions of the CWI pathway in cell-wall biogenesis and morphogenesis are very well-known [3]. More recently, other authors have revised other functions of the CWI beyond the cell wall [7,8]. In this review, we focus on the direct mechanisms by which the CWI pathway affects the cell-cycle control system, considering the latest advances made.

## 2. The CWI Pathway and Start

The G1/S transition, called Start in budding yeast, is a fundamental point in cell-cycle control, in which cells irreversibly commit to a new round of cell division [9]. Start comprises the activation of a transcriptional program which involves the periodic expression of hundreds of genes, including genes coding for cell-cycle regulators and the cellular machinery needed for the S-phase or cell-wall biosynthesis. Two transcription factors—SBF and MBF—regulate this transcriptional wave at G1/S [10]. Both are heterodimeric proteins composed of a DNA-binding factor, Swi4 (SBF) or Mbp1 (MBF), and a common protein, Swi6. In the case of SBF, it has been characterized that Swi6 acts, in part, by alleviating an auto-inhibitory intramolecular interaction in Swi4 that blocks its DNA-binding domain [11]. Although SBF and MBF preferentially regulate a subset of specific genes, there is an important functional redundancy between them [12,13,14]. The importance of the Start transcriptional program has been evidenced by the lethality of *swi4 swi6* and *swi4 mbp1* double mutants [15]. Strikingly, the *mbp1 swi6* cells are viable, indicating that Swi4 could activate transcription in the absence of Swi6 [16,17,18].

The CWI pathway has been related to SBF at multiple levels. However, we still lack a complete picture of how it directly regulates the Start transcriptional machinery and the implications that this has for the control of cell-cycle entry. 

### 2.1. SBF and Slt2: Co-Regulation of Cell-Wall Gene Transcription

Genetic and biochemical studies have long connected the CWI pathway with the transcription factor SBF, supporting the idea that SBF is involved in the maintenance of cell integrity. First, the cell lysis defect of an *slt2∆* mutant was suppressed by *SWI4* overexpression [19], and mutations in both Slt2 and Pkc1 showed synthetic lethality with Swi4 and Swi6 mutations [20]. Second, *swi4∆* and *swi6∆* mutants have been shown to be hypersensitive to the cell-wall-perturbing compounds CFW and SDS [20], and the lethality of *swi4∆* in some backgrounds was suppressed by an osmotic stabilizer [19]. In addition, SBF is responsible for the periodic expression (peaking in the G1/S transition) of cell-wall protein genes, genes which are also regulated by the CWI pathway [20] through its main transcriptional effector, Rlm1 [21]. Interestingly, inactivation of SBF eliminated cell-cycle periodic expression, but did not affect the global mRNA level, whereas CWI pathway mutants showed reduced expression of cell-wall genes without affecting their periodic expression [20]. Therefore, there exists a co-regulation of cell-wall genes, in which the CWI pathway and the cell-cycle transcription factor SBF act in parallel: Slt2 activates Rlm1 to control the global level of expression, whereas SBF modulates this expression throughout the cell cycle. Genomic studies have further confirmed the existence of a group of genes that are coregulated by both SBF and Slt2 [22].

In addition to collaborate in the expression of common target genes, direct relationships between SBF and the CWI pathway have been described. Biochemical studies have shown that Slt2 phosphorylates Swi6 and Swi4 and associates with SBF [19,22]. In fact, it has been proposed that, in the absence of Swi6, Slt2 could control the association of Swi4 with the promoters of a subset of specific genes [22]. Subsequent studies have described that, in response to cell-wall stress, Slt2 and its pseudokinase paralog Mlp1 activate SBF through a noncatalytic mechanism [23,24,25]. In this mechanism, Slt2 and Mlp1 must be in their active form (phosphorylated), but their kinase activity is not required. Then, Slt2 and Mlp1, in complex with Swi4, associate with the *FKS2* promoter independently of Swi6 [24]. Slt2 associates through a D-motif binding site common in MAPK with the D-domain present in the C-terminal autoinhibitory region of Swi4. As a consequence, the Swi4 DNA-binding domain is released to direct promoter association [25]. Although, in this mechanism, Slt2 replaces Swi6 to allow for Swi4 binding to DNA, Swi6 must be incorporated into the Slt2–Swi4 complex to allow subsequent recruitment of RNA pol II and activation of *FKS2* transcription in response to cell-wall stress [24]. 

Remarkably, this mechanism is not general for SBF-regulated genes. On the contrary, it only affects a very small number of genes: *FKS2*, *CHA1*, *YLR042c*, and *YKR013* [23,24]. Note that the *FKS2* gene is an exceptional cell-wall gene, as it is only expressed under cell-wall stress conditions and its expression—although Slt2-dependent—is not mediated by Rlm1, as it occurs for the vast majority of cell-wall genes [21], but by the SBF transcription factor [24]. In spite of this, the expression of *FKS2* by SBF is not cell-cycle regulated [26]. Interestingly, in the case of *YKR013w*, this mechanism acts specifically through only one of the three SBF binding sites present in its promoter, the other sites likely being responsible for the periodic expression of the gene at Start [23]. Moreover, although Slt2 and Swi6 both alleviate the Swi4 self-inhibition mechanism for DNA binding, they act through specific and distinct interactions with Swi4, as preventing the interaction of Slt2 with Swi4—which inhibits *FKS2* expression—does not prevent either the association of Swi4 with Swi6, nor SBF-dependent activation of cell-cycle genes [25]. In conclusion, this direct regulation of Swi4 by Slt2 is independent of the role of SBF in cell-cycle-regulated transcription. How this mechanism of Swi4 activation by Slt2 is restricted to a handful of SBF-regulated genes (and SBF binding sites) remains an open question.

### 2.2. Regulation of the Subcellular Localization of Swi6 by Slt2

In addition to the connections mentioned above, two other mechanisms involving the direct regulation of mediators of the Start transcriptional program by the CWI pathway have functional implications for cell-cycle control. One of them involves controlling the subcellular localization of the transcription factor Swi6. The localization of Swi6 changes throughout the cell cycle: it is nuclear in G1, relocates to the cytosol at G2/M, and re-enters the nucleus at the end of mitosis [27,28]. This regulation along the cell cycle is controlled by the phosphorylation of Ser^160^ by Clb6-Cdc28 kinase [29], which acts by inhibiting a nearby nuclear localization signal (NLS) [30]. Later, it was identified that Swi6 localization also undergoes changes in response to cell-wall stress and that these changes are mediated by Slt2. Under cell-wall stress conditions, there is a transient accumulation of Swi6 inside the nucleus: it rapidly enters the nucleus through its interaction with the Slt2-Swi4 complex discussed above, and then relocates to the cytosol due to phosphorylation of Ser^238^ by Slt2 [31]. This phosphorylation acts by inhibiting a nearby second NLS present in Swi6. As expected, this phosphorylation and the consequent nuclear import blockage has an inhibitory function on the sustained expression of *FKS2* under cell-wall stress conditions [31]. Thus, Swi6 has one NLS that is regulated throughout the cell cycle by Cdc28, and another NLS which is regulated in response to cell-wall stress by Slt2. Slt2-dependent phosphorylation can be detected by Western blot, as the appearance of slow migrating bands [31]. It is interesting to note that this band-shift has also been observed—although to a lesser extent—under basal conditions ([19] and our unpublished results), suggesting that regulation of the subcellular localization of Swi6 by Slt2 may have implications beyond cell-wall stress conditions. In support of this, inactivation of the Slt2-regulated NLS of Swi6 also reduced the expression of the *CLN2*: *lacZ* gene under basal conditions [31]. It is foreseeable that this mechanism could have a more global impact, affecting other genes regulated by Swi6. In short, the MAPK Slt2, by phosphorylating Swi6 to restrain its nuclear import, may act as an inhibitor of the Start transcriptional program. However, this does not seem to be a limiting step in the case of *CLN2* and *CLN1* genes, as inactivation of Slt2 does not affect their expression [19]. Global expression analysis with an Swi6 mutant protein lacking the Slt2-regulated NLS may help in identifying target genes whose expression is affected by this regulatory mechanism, both in basal and cell-wall stress conditions. 

### 2.3. Regulation of the Start Repressor Whi7 by the CWI Pathway

A new direct mechanism that links the CWI pathway with Start has been revealed with the discovery of the Start repressor Whi7. Although SBF is bound to the G1/S gene promoters in early G1, Start transcription is inhibited by the association of transcriptional repressors Whi5 and Whi7, which are functional homologs of Rb in mammalian cells [32,33,34]. Whi7 also negatively regulates Start through the retention of the Cln3 cyclin in the endoplasmic reticulum membrane [35]. Whi5 and Whi7 show sequence homology and similar cell-cycle-dependent localization, but Whi7 is a less-abundant protein than Whi5, which could explain why Whi5 is the main transcriptional repressor under normal conditions [32,36]. Moreover, both repressors show different preferences for promoter binding: Whi5 is preferentially associated with the G1/S cyclin genes *CLN1* and *CLN2*—key regulators of Start—whereas Whi7 has a preferred association with cell-wall genes [36]; however, Whi7 could substitute for Whi5 in the control of Start when overexpressed [32].

*WHI7* expression is upregulated under various stress conditions [37,38,39]. Under cell-wall stress, an eightfold induction of *WHI7* transcription has been observed. This induction is mediated by Slt2 and the Rlm1 transcription factor, which binds to the *WHI7* promoter [40,41]. Strikingly, inactivation of Slt2 or Rlm1 not only affects cell-wall stress induction, but *WHI7* mRNA levels are also almost depleted under both normal and stress conditions [36]. Consistently, Whi7 protein levels were also decreased in a *pck1* mutant. Thus, *WHI7* is a gene whose expression is totally dependent on the CWI pathway; in particular, on the MAPK Slt2 and Rlm1. This suggests that Whi7 must be an important mediator of CWI pathway functions, specifically in connection with cell-cycle regulation.

As mentioned above, in normal growth conditions, Whi7 plays a modest role as a Start repressor, compared to Whi5. Importantly, this situation changes under cell-wall stress conditions. In response to this stress, *cln3* mutant cells arrest in G1. This arrest is mediated mostly by Whi7 and not by Whi5, as *whi7* deletion is more efficient than *whi5* deletion to override the Start blockage. Similarly, Whi7 and not Whi5 mediates the partial G1 arrest observed in wild-type cells. Thus, under cell-wall stress conditions, Whi7, and not Whi5, becomes the main Start repressor to regulate cell-cycle entry [36]. 

The gain-of-function of Whi7 under cell-wall stress conditions is totally coherent with the fact that *WHI7* expression is dependent on the CWI integrity pathway and the preference of Whi7, compared to Whi5, for binding to cell-wall gene promoters. Is this gain-of-function simply the result of the stress-dependent accumulation of Whi7 protein? The fact that Whi7 protein levels become similar to those of Whi5 protein (which remain constant during stress) after cell-wall stress treatments provides evidence against a merely quantitative explanation. Alternatively, qualitative differences between the Start repressors probably explain their different abilities to inhibit cell-cycle progression after cell-wall stress. It is possible that, in response to stress, the CWI pathway could increase the intrinsic functionality of Whi7. Furthermore, it is interesting to remark that each repressor has different determinants for binding to promoters. Whi5 binding relies on an integral SBF complex, as binding to any of the Swi4 or Swi6 proteins, as well as association to promoters, is totally destroyed in the absence of the other member of the complex [33,34]. On the contrary, Whi7 can associate with promoters and interact with monomeric Swi4 in the absence of Swi6 [36]. The inhibition of Swi6 import in response to cell-wall stress may result in an enrichment of monomeric Swi4 in SBF-regulated promoters, making regulation more dependent on Whi7 than Whi5. The functional specialization of Whi7 in stress conditions opens the possibility that stress-specific Start repressors may be key elements linking signaling pathways with cell-cycle arrest and subsequent cell-cycle re-entry under adverse conditions.

### 2.4. The CW/START (Cell-Wall/Start) Checkpoint

A new link between the CWI pathway and the G1/S control, called the cell-wall/Start checkpoint (CW/START) or the Rlm1-dependent checkpoint, has been recently described [42]. The authors noticed that *rlm1* mutant cells growing in a nonfermentable carbon source with low osmolarity underwent the formation of cell aggregates composed of a large mother cell surrounded by several small satellite daughter cells. These aggregates formed when the mother cell progressed through several rounds of cell division, whereas the daughter cells were unviable due to defects in cell integrity and the actin cytoskeleton appearing after cytokinesis, causing their rapid shrinkage and death. In such culture conditions, a strong activation of the CWI pathway, including Rlm1, occurs. This activation is consistent with cell-wall defects and the role of Rlm1-mediated transcription of cell-wall biosynthesis genes in the repair of the damage. It is worth noting that phenotypic analysis of the mother cells revealed that Rlm1 plays another fundamental role as, in its absence, the G1 phase was significantly shortened compared to that of wild-type cells. This observation led the authors to propose that Rlm1 negatively regulates Start when grown under a nonfermentable carbon source and low osmolarity conditions. This environment-specific G1 delay would act as a checkpoint that responds to cellular damage blocking Start, thus allowing for optimal biosynthesis of the cell wall in the poor media before embarking on a new division, in order to guarantee the viability of the future daughter cell.

The mechanism by which Rlm1 regulates Start is unknown, but it does not involve inhibition of the CDK Cdc28 by Tyr^19^ phosphorylation [42]. A good possibility is that Rlm1 might be acting through Whi7, which, as commented above, is a repressor of the Start transcriptional program whose expression depends on Rlm1 [36]. However, *slt2* mutant cells, which also showed depletion of the Whi7 protein [36], lacked the satellite daughter phenotype observed in *rlm1* cells [42]. This is intriguing, given that the activation of Rlm1 is dependent on Slt2, although it cannot be ruled out that, under these particular conditions, Rlm1 can be activated by mechanisms independent of Slt2 and that Whi7 could be involved.

More work is needed to unveil the exact nature of the cellular damage caused by respiration under low osmolarity conditions that triggers this checkpoint, and how that signal reaches Rlm1 in an Slt2-independent way, as well as the Rlm1 targets delaying the G1/S transition.

## 3. The CWI Pathway and DNA Replication

### 3.1. The CWI Pathway Regulates the CDK Inhibitor Sic1

A key event in the G1/S transition of the cell cycle is degradation of the CDK inhibitor Sic1. Sic1 blocks the activity of the S-phase CDKs Clb5,6-Cdc28, preventing the activation of DNA replication [43,44]. When the Start transcriptional program is activated, the G1/S CDKs Cln1,2-Cdc28 accumulate and phosphorylate Sic1 at multiple sites, targeting it for degradation mediated by the ubiquitin ligase SCF^Cdc4^ [45,46]. Clb5,6-Cdc28 further phosphorylate Sic1, establishing positive feedback that ensures their abrupt and irreversible activation, and the consequent initiation of DNA replication [47].

The TORC1 complex is the major controller of growth in eukaryotes. The inhibition of TORC1 by rapamycin, which mimics a nutrient deficiency, causes a blockage of cell-cycle progression in G1. This arrest is due—in addition to the decrease in the cellular levels of Cln cyclins—to the increase in Sic1 protein levels. This increase occurs through stabilization of the Sic1 protein caused by its phosphorylation at the Ser^173^ residue [48]. The kinase responsible for such phosphorylation is the MAPK Slt2 of the CWI pathway, this phosphorylation being reversed by the phosphatase PP2A^Cdc55^ [49]. Under normal conditions, TORC1 indirectly inhibits Slt2 [50,51] and activates phosphatase PP2A^Cdc55^ by downregulation of the inhibitory Greatwall (Rim15) endosulfin (Igo1,2) pathway. This maintains Sic1 Ser^173^ in a dephosphorylated state. However, when TORC1 is downregulated by rapamycin or an absence of nutrients, Slt2 kinase activity is induced and the Greatwall–endosulfin pathway is activated to inhibit PP2A^Cdc55^, resulting in the phosphorylation of Ser^173^ and consequent stabilization of Sic1 [49,52]. Accordingly, an *slt2* mutant showed defects in the rapamycin-induced G1 arrest.

The phosphorylation of Sic1 in Ser^173^ could be also relevant in cell-cycle control in proliferating cells under normal conditions. It has been proposed that Rim15 is negatively regulated by Cln-Cdc28 [52,53]. This could facilitate Ser^173^ dephosphorylation by PP2A^Cdc55^, contributing to Sic1 degradation at the G1/S transition. Some results have shown that cells expressing the Sic1^T173A^ mutant protein slightly advance the accumulation of CDK activity [52] and DNA replication [54] after release from an α-factor arrest in the absence of stress. To the best of our knowledge, no effect in DNA replication initiation has been associated with Slt2 inactivation under normal growth conditions, calling into question whether this mechanism—or, at least, the role of Slt2—is critical for the control of Start under optimal conditions. Note that another MAPK, Hog1, phosphorylates Sic1 Ser^173^ to impose a G1 arrest in response to hyperosmotic stress [54]. In any case, by stabilizing Sic1, Slt2 plays an inhibitory role in the initiation of DNA replication, acting as a brake for the G1/S transition when nutrient availability is not adequate.

### 3.2. The PM/CW (Plasma Membrane/Cell Wall) Damage Checkpoint

The existence of a cell-cycle checkpoint by which cells block the initiation of DNA replication in response to PM/CW damage caused by SDS has been suggested [55]. Two mechanisms that contribute to this response have been characterized. The first one plays a more important role in cell-cycle arrest and involves degradation of the Cdc6 protein. Cdc6 is an essential protein required for DNA replication, which acts in the formation of pre-replicative complexes at origins [56]. Its degradation in response to cell-surface damage is mediated through phosphorylation by the yeast GSK-3 Mck1 at Thr^39^ and Thr^368^. The second mechanism involves stabilization of the CDK-inhibitor Sic1. In fact, S-phase CDKs’ activity is greatly reduced after SDS treatment [55]. These two processes help to block DNA replication and ensure cell survival while membrane damage is under repair.

It is not surprising that the CWI pathway is directly involved in plasma membrane damage repair, through reorganization of the actin cytoskeleton from the growth point toward the area of damage [57] and the transcriptional activation of genes involved in cell-wall biosynthesis [21]. However, more importantly, different observations strongly suggest that the CWI pathway also plays an important role in this checkpoint response. The molecular mechanism underlying Sic1 stabilization in this condition has not been described. However, it is very likely that, as occurs with nutritional [48] or osmotic stress [54], Sic1 is stabilized by phosphorylation in Ser^173^. Taking into account that treatment with SDS activates the CWI pathway, Slt2 might be the kinase responsible for this phosphorylation, as occurs in the response to nutritional stress. Regarding Cdc6 degradation, phosphorylation of Thr^368^ by Mck1 is dependent on a priming phosphorylation at nearby Ser^372^ by Cdc28 [58]; however, it has been suggested that Cdc28 kinase is not required for Cdc6 degradation in response to plasma membrane damage, and that other kinases can replace Cdc28 to prime Cdc6 phosphorylation by Mck1 in this condition [55]. Slt2 is a good candidate as the phosphorylation sites of CDK and MAPK are similar and, as indicated, Slt2 is activated under this condition. On the other hand, Slt2 positively regulates Mck1 in Ca^2+^ signaling [59]. Thus, it is possible that the CWI pathway could also regulate the degradation of Cdc6 in response to SDS. 

It is interesting to note that the cell-cycle arrest after SDS treatment is only partially alleviated by a nonphosphorylatable Cdc6 protein, or is not affected by *sic1* deletion [55]. This could be due to the redundancy of both mechanisms or, alternatively, may point to the existence of additional mechanisms acting to restrain the G1/S transition in response to SDS treatment.

## 4. The CWI Pathway and the G2/M Transcriptional Program

### 4.1. Pkc1 Negatively Regulates the G2/M Transcriptional Program

The cell-cycle-regulated transcriptional wave that occurs in late G2 involves a large variety of genes encoding proteins important for mitosis, among them the gene of the major mitotic cyclin, *CLB2*. This transcriptional program is regulated by the transcriptional complex constituted by Mcm1–Fkh2–Ndd1 [10]. The treatment of cells with the Pkc1 inhibitor cercosporamide resulted in increased and advanced expression of the *CLB2* gene cluster, suggesting that Pkc1 could act to negatively regulate this transcriptional program. Pkc1 associates with the promoter of genes of the *CLB2* cluster in a cell-cycle-regulated manner: it is recruited to target promoters during S-phase, while binding declines as cells progress into G2/M when the transcriptional program is turned on. In vitro kinase assays have demonstrated that Pkc1 phosphorylates Ndd1 in two residues: Ser^520^ and Ser^527^. Interestingly, mutation of these Ser residues to Ala caused an increased and advanced association of Ndd1 to promoters of the *CLB2* cluster genes, and a higher level of gene expression was consistently detected [60]. All of these observations are consistent with Pkc1 inhibiting the expression of *CLB2* cluster genes by phosphorylating Ndd1 transcription factor. Cells expressing the Ndd1^S520A,S527A^ mutant protein progressed through the G2/M transition faster after release from an HU-induced arrest and had severe growth defects in the presence of HU or at high temperature [60]. Thus, Pkc1-dependent phosphorylation of Ndd1 must be important for cell physiology under certain conditions. 

### 4.2. The CWI (Cell-Wall Integrity) Checkpoint

The study of mutant strains in enzymes involved in the biosynthesis of cell-wall components have revealed that, in addition to the expected defect in bud growth, there is a blockage in cell-cycle progression at the G2 phase: cells accumulate with replicated DNA and duplicated but not separated SPB. These observations have led to the proposal of the existence of the CWI checkpoint by which cells respond to alterations in cell-wall structure by inhibiting entry into mitosis [61]. This arrest is associated with an inhibition of mitotic CDK activity caused by a drop in the levels of the major mitotic cyclin, Clb2. In fact, *CLB2* overexpression partially alleviates the cell-cycle arrest induced by cell-wall damage.

The regulation of Clb2 by the CWI checkpoint occurs at the transcriptional level. As commented above, the cell-cycle transcriptional wave in G2 phase that includes the *CLB2* gene is controlled by the Mcm1–Fkh2–Ndd1 transcription complex. Increased levels of Fkh2 [61] and Ndd1 [62] override the checkpoint, indicating that inhibition of Fkh2–Ndd1 mediated transcription is critical for the induced arrest. However, the regulation of Fkh2–Ndd1 by cell-wall damage is not a direct event; rather, the CWI checkpoint regulates another cell-cycle transcription factor: Hcm1. Hcm1 is responsible for the cell-cycle transcriptional wave in S-phase, in which many genes involved in spindle assembly and dynamics, chromosome segregation, or budding are expressed, including the genes for the transcriptional factors *FKH2* and *NDD1* required for the next transcriptional wave [63]. When the CWI checkpoint is triggered by cell-wall damage, the cellular levels of Hcm1 drop dramatically. This regulation occurs through a post-transcriptional mechanism involving the phosphorylation of at least Ser^61^, Ser^65^, and Ser^66^, which likely destabilizes the protein. As a consequence, the transcription of many genes regulated by Hcm1 (including the *FKH2* and *NDD1* genes) is impaired. This leads to inhibition of transcription in G2, resulting in cell-cycle arrest [64]. In this context, the inhibition of Ndd1 by Pkc1 phosphorylation described above is expected to contribute to the checkpoint arrest.

The fact that conditions that activate the CWI checkpoint also activate the CWI pathway, and that these Hcm1 phosphorylation sites are Ser–Pro sites, suggest that the MAPK Slt2 could be involved in these phosphorylation events. In support of this hypothesis, Hcm1 protein levels increase when Slt2 is inactivated and decrease when Slt2 is constitutively activated. Importantly, in the absence of Slt2, Hcm1 is partially stabilized, similar to that observed in the nonphosphorylatable Hcm1 mutant [64]. All these observations have led to the suggestion that MAPK Slt2 triggers Hcm1 degradation by phosphorylation at Ser^61^, Ser^65^, and Ser^66^ in response to cell-wall damage, eventually inducing the blockage of cell-cycle progression in G2/M. Supporting this, more recently, it has been reported that mutant strains in all components of the MAPK cascade are defective in blocking cell-cycle progression, as deduced from the presence of SPB separation under cell-wall stress conditions [65]. Future work is necessary to demonstrate whether Hcm1 is indeed a substrate for Slt2 kinase, and to evaluate the relevance of this regulation in the response to cell-wall damage.

## 5. The CWI Pathway and Mitosis

The key step in mitotic entry is the activation of mitotic CDK through the removal of inhibitory phosphorylation in a specific Tyr residue (Tyr^19^ of Cdc28 in *S. cerevisiae*). This phosphorylation is controlled by the Wee1 kinase (Swe1 in *S. cerevisiae*) and the Cdc25 phosphatase (Mih1 in *S. cerevisiae*). A sophisticated system-level mechanism for the control of mitotic entry in both yeast and mammals ensures step-wise activation of CDK: it is first activated at a low level, in order to enter mitosis, and then strongly activated to progress through mitosis [66,67,68]. Both Swe1 and Mih1 are hyperphosphorylated at multiple sites in a cell-cycle-regulated manner. Cdc28 partially phosphorylates Swe1 to activate it, thus limiting Clb2-Cdc28 activity to a basal level to initiate entry into mitosis [67,69]. Subsequently, complete hyperphosphorylation of Swe1 by Cdc28 and other kinases will inactivate Swe1, inducing its degradation and, thus, allowing the burst of Cdc28 activity necessary for mitotic progression [69,70,71]. On the other hand, Mih1 is hyperphosphorylated at the beginning of the cell cycle by casein kinase I. Although the role of this phosphorylation is not clear, it has been suggested that it may have an inhibitory role [72]. In mitotic entry, Mih1 is partially dephosphorylated and undergoes activating phosphorylation by Cdc28 [73]. As can be seen, the regulation of Swe1 and Mih1 is complex, with both proteins undertaking both inhibitory and activating phosphorylations.

### 5.1. Pkc1 Negatively Regulates the Mih1/Cdc25 Phosphatase

In cells that contains a single mitotic cyclin, Clb2, Mih1 phosphatase is essential for entry into mitosis at elevated temperature [74]. The Mih1 sequence presents four possible Pkc1 phosphorylation consensus sites. Substitution of these Ser by Asp or Glu (Mih1^3E1D^) reduces the ability of Mih1 to trigger mitotic entry. This is due to the nuclear exclusion of the protein, as evidenced by the fact that the functionality of Mih1^3E1D^ is recovered by forcing its nuclear localization with the SV40 NLS. It could be considered that this mechanism prevents Mih1 from activating Cdc28 by dephosphorylating Tyr^19^. However, the suppression of the mitotic arrest of the *clb1,3,4 mih1* strain by Cdc28^Y19F^ is very slight, and the difference in the extent of Tyr^19^ phosphorylation between Mih1^3E1D^ and Mih1 is very small, indicating that the blockage of mitotic entry associated with Mih1 inactivation involves other targets besides Cdc28.

Two observations suggest that Pkc1 is the kinase responsible for the phosphorylation of these four Ser residues: first, bands of lower electrophoretic mobility associated with in vivo phosphorylation of these residues (absent in Mih1^4A^, a mutant protein with the four Ser substituted by Ala) disappear when Pkc1 is inactivated in the cell. Second, *PKC1* overexpression causes Mih1 nuclear exclusion, but has no effect on the Mih1^4A^ mutant protein. These observations suggest that Mih1 is downregulated by Pkc1-dependent phosphorylation, which excludes Mih1 from the nucleus under certain conditions. An attractive hypothesis is that this mechanism could help to prevent premature entry into mitosis until bud formation is complete, as the CWI pathway is activated during the budding process and inactivated in the G2/M transition [75,76,77]. However, there is no definitive evidence that Mih1 is a substrate for Pkc1 and, in fact, other authors have reported that Pkc1 does not phosphorylate Mih1 in vitro [73]. Therefore, it is necessary to clarify the role of Pkc1 in the phosphorylation of Mih1 in further research.

### 5.2. The Morphogenesis Checkpoint

The morphogenesis checkpoint has been proposed as a mechanism that blocks the cell cycle in G2/M in response to defects in the actin cytoskeleton caused by various stresses, allowing time for bud formation and growth before cell division [78]. More recently, it has been proposed that the signal that activates the checkpoint is actually a defect in membrane trafficking at sites of polarized growth [79], and that this mechanism could function as a cell sizer to block the G2/M transition until the bud has grown sufficiently [80], although the latter is a matter of controversy [81].

Blockage of cell-cycle progression occurs when Cdc28 is inhibited by phosphorylation of Tyr^19^, a mechanism regulated by Swe1 tyrosine kinase and Mih1 phosphatase. In the morphogenesis checkpoint, in response to budding defects or alterations of the actin cytoskeleton, Swe1 degradation is deregulated and the protein accumulates to inhibit Cdc28 [78,82]. Stabilization of Swe1 is insufficient to generate a long mitotic delay, but simultaneous inactivation of Mih1 can cause a lethal G2/M blockage. These observations have led to the suggestion that the morphogenesis checkpoint operates both through Swe1 stabilization and Mih1 inhibition [78,83].

The CWI pathway plays an important role in the morphogenesis checkpoint. Mutants in Rho1, Pkc1, and the MAPK Slt2 cascade have been reported to override the G2/M arrest [84]. High Ca^2+^ concentrations also induce Slt2-dependent G2/M arrest [85]. Specifically, acting through GSK-3 Mrc1, Slt2 downregulates Hsl1 kinase [59], which is involved in the control of Swe1 degradation [83]. This mechanism may also be acting in response to disturbances in the actin cytoskeleton. Furthermore, as Slt2 inactivation caused checkpoint defects in an *hsl1* mutant but not in a *mih1* mutant strain, it has been suggested that Slt2 could act by inhibiting Mih1 phosphatase [84].

On the other hand, in the context of the studies connecting bud growth with the control of mitosis, Pkc1 phosphorylates the endosulphin proteins Igo1/Igo2 and Cdc55, in order to prevent their interaction and the consequent inhibition of the PP2A^Cdc55^ phosphatase [86]. PP2A^Cdc55^ regulates the phosphorylation state of both Swe1 and Mih1, playing an ambivalent role in mitosis. PP2A^Cdc55^ partially dephosphorylates Mih1 and counteracts the activating phosphorylation of Swe1 by Cdc28 upon entering mitosis [67,73]. Thus, PP2A^Cdc55^ plays a positive role in mitotic entry, by helping to establish a basal threshold for Cdc28 activity. Therefore, mutants in PP2A^Cdc55^ show a significantly delayed entry into mitosis, inhibiting Cdc28 through phosphorylation at Tyr^19^ [73]. However, in addition to this positive role, PP2A^Cdc55^ negatively regulates mitotic progression. Various studies have indicated that PP2A^Cdc55^ counteracts Cdc28-dependent phosphorylation in many substrates [87,88], which implies a general function as an opposing activity to CDKs in mitosis—a role well-characterized in other systems [89]. Furthermore, PP2A^Cdc55^, by reverting Cdc28-dependent phosphorylation of Swe1, could avoid the full hyperphosphorylation of Swe1 necessary for its degradation, thus negatively regulating the full activation of Cdc28 in mitosis. In fact, the progression through mitosis depends on PP2A^Cdc55^ inactivation [90]. More important, PP2A^Cdc55^ activity is necessary to inhibit mitotic progress in response to morphogenetic defects that activate the checkpoint [91]. In this context, the activation of PP2A^Cdc55^ by Pkc1 mentioned above may reflect the role of Pkc1 in retaining mitosis through the morphogenesis checkpoint. As PP2A^Cdc55^ has also a positive role in initiating mitosis, it is tempting to speculate that the abovementioned mechanism of Pkc1 downregulating Mih1 might counteract this positive role of PP2A^Cdc55^ when Pkc1 is activated. In fact, restricting the nuclear import of Mih1 causes an extended delay during the morphogenesis checkpoint response [92]. The mutation of Pkc1-phosphorylation sites on Cdc55 and Igo2 had no effect in mitotic progression in nonstressed cells [86]. It would be interesting to test the physiological relevance of Pkc1-dependent regulation of PP2A^Cdc55^ when the morphogenesis checkpoint is in action.

## 6. A Collection of Checkpoints Related to Cell-Surface Perturbations: A Unified Vision

Checkpoints are surveillance mechanisms that prevent the initiation of a cell-cycle process if abnormalities exist, or when a previous process has not been successfully completed. As described above, in recent years, various checkpoints have been proposed to stop cell-cycle progression in response to different perturbations affecting the cell surface, in which the CWI pathway has been involved. Furthermore, additional mechanisms have been described that support a negative role for the CWI pathway in cell-cycle progression, which are clearly complementary or overlapping with some of the proposed checkpoints, although they have not been named as such.

We propose a unified vision of all these checkpoints and mechanisms. For example, the DNA damage checkpoint responds to different signals depending on the damage, and the DNA replication checkpoint responds to problems in replicative forks. These responses affect different cell-cycle regulators, depending on the stage in the cycle. Strikingly, all of these responses converge through a common core mechanism, composed of a signaling pathway that involves apical kinases ATR/ATM and effector kinases Chk1/Chk2 [93]. Similarly, the various proposed checkpoints related to cell-surface alterations may respond to different signals: actin cytoskeletal alterations or membrane trafficking (morphogenesis checkpoint), alteration of the plasma membrane by SDS (PM/CW checkpoint), or defects in cell-wall biosynthesis (CWI checkpoint or CW/START checkpoint); however, all of them converge in the activation of the CWI pathway, which acts on different effectors to block the cell cycle at different stages (Figure 1). Thus, the CWI pathway emerges as a major brake in the control of the cell cycle when the cell surface is compromised.

The differences observed between the different checkpoints could be due to various causes. A first possibility is that the different treatments used are sensed by different sensors generating different outputs. For instance, in the case of lesions due to *fks1* mutation (CWI checkpoint), the damage is detected by the Sho1-branch of the HOG pathway, which activates Hog1 and, acting through dynactin and the Las17 complex, transduces the signal to the MAPK Slt2 cascade, without requiring the classical Wsc1-3, Mid2, and Mtl1 sensors of the CWI pathway [65]. Wsc1-3 and Mid2 also do not participate in the morphogenesis checkpoint [84]. Classical pathway sensors can act on other stresses, but it has been described that compounds that damage the cell wall use pathway-specific sensors [21]. In this sense, it would be interesting to characterize the molecular basis of the signals generated under different conditions, the components required to transduce them to the CWI pathway, and at which level in the pathway the signal impinges. 

Another alternative is that the response would depend on the intensity of the generated signal. For instance, Hog1 activation is induced by cell-wall perturbation in the CWI checkpoint, but well-known downstream effectors of Hog1 are not activated. This is probably due to the fact that Hog1 activation by cell-wall perturbation is very weak, when compared to Hog1 activation by hyperosmotic stress [65]. Similarly, it is possible that, depending on the intensity of the signal generated by a specific cell-surface damage, the response is directed to one or another cell-cycle effector. In this sense, it would be interesting to compare, in parallel, the intensities of the MAPK Slt2 activation in response to the treatments used in the different checkpoints, in order to evaluate this possibility.

Finally, it cannot be ruled out that, in some cases, the differences were caused by the different experimental approaches used to generate the damage. For example, using G1-synchronized cells, it has been reported that SDS (PM/CW checkpoint) blocks DNA replication, while *fks1* mutation (CWI checkpoint) blocks mitosis. This could reflect that the SDS-induced damage was sensed earlier than *fks1*-induced damage. Maybe the inactivation of glucan synthase requires time to generate damage in the cell-wall structure and this damage occurs in the period of bud growth, once the cells have passed Start and activated DNA replication. It would be interesting to carry out an experiment with SDS on post-replicative cells, in order to see whether SDS acts on the G2/M transition or on the next Start, and an experiment with the *fks1* mutation maintaining cells in G1 for enough time to generate cell-wall damage before Start, in order to see whether it could act on DNA replication. Or, is there a homogeneous arrested phenotype when the stress is induced in asynchronous cultures?

It can be expected that future works focused on the connections between the CWI pathway and the control of cell proliferation in response to alterations in the cell surface and morphogenesis will clarify all of these aspects.

## 7. The CWI Pathway in the Response to DNA Damage

Genetic material is constantly exposed to insults caused by physiological processes, such as replication errors, defective activities of enzymes or reactive oxygen species, or by external physical and chemical agents. To cope with these threats, cells have developed surveillance mechanisms—namely, DNA damage and replication checkpoints, collectively referred to as the DNA integrity checkpoint hereafter—to arrest cell-cycle progression and initiate damage repair [93,94]. The DNA integrity checkpoint has evolved to become a complex signaling network which is evolutionarily well-conserved. The major regulators of this network are apical kinases Mec1 and Tel1 (ATR and ATM in mammals) that, through the adaptor proteins Rad9 and Mrc1 (53BP1 and claspin in mammals), activate the downstream effector kinases Rad53 (Chk2 in mammals) and Chk1. Once activated, DNA damage signaling kinases mediate hallmark responses, such as cell-cycle arrest, inhibition of origin firing, protection and restart of stalled replication forks, induction of a transcriptional response, induction of DNA repair, control of dNTP levels, and the induction of apoptosis (in vertebrates) [93,95].

Several studies have provided evidence supporting the existence of cross-talk between the DNA integrity checkpoint and cellular morphogenesis in budding yeast. DNA integrity checkpoint mutants have aberrant cell morphology, cell wall, and polarized growth [96,97,98], and there exist proteins that have separable functions in both the cell wall and genome integrity pathways [98,99]. Therefore, it is not surprising that the CWI pathway has been related to the DNA integrity checkpoint in the last decade.

### 7.1. Slt2 and the DNA Damage Response

First reports connecting Slt2 and the DNA damage response described synthetic genetic interactions between mutations in *SLT2* and DNA damage genes [100,101]. More important, Slt2 is activated in response to different genotoxic stresses, such as treatment with the mutagen methylmetanosulfonate (MMS), hydroxyurea (HU), phleomycin, or UV irradiation [102], suggesting that Slt2 plays an important role in the cellular response to DNA damage. Slt2 activation involves the phosphatase Msg5, which dephosphorylates Slt2 to maintain it in a low-activity state in the absence of stress. When DNA damage occurs, Msg5 is degraded, leading to Slt2 activation. Although components of the CKI pathway upstream of Slt2 are not directly involved in the mechanism by which DNA damage activates Slt2, a functional pathway is needed. The authors propose that the pathway must have a basal level of activity that will be modulated in response to DNA damage directly at the level of Slt2 by Msg5 [103]. 

On the other hand, Slt2 is a direct substrate for checkpoint kinases. A genomic study has indicated that Slt2 is phosphorylated at Ser^423^ and Ser^428^ in response to MMS [104]. These same sites were phosphorylated by Rad53 (Ser^423^) and Mec1/Tel1 (Ser^428^) when cells were treated with caffeine [25]. As described above, the phosphorylation of Ser^423^ blocks the interaction of Slt2 with Swi4. Although the functional relevance of this phosphorylation in the context of DNA damage is unknown, it has been suggested that it could help redirect Slt2 to targets involved in the DNA damage response. 

What are the targets of Slt2 in response to genotoxic stress? A protein candidate to mediate the role of Slt2 in response to DNA damage is Swe1. A morphogenetic function of the DNA integrity checkpoint consists of switching from apical to isotropic bud growth after DNA damage by the degradation of Swe1, mediated by Rad53 [96]. The *slt2* mutant cells have shown hyperpolarized bud morphology and defects in Swe1 degradation under replicative stress, indicating that DNA-damage-induced Swe1 degradation is mediated by Slt2 [102]. This role is apparently contradictory to the results previously described in the morphogenesis checkpoint context as, in that case, Slt2 is supposed to act by stabilizing Swe1. However, it must be taken into account that Swe1 also acts as a downstream effector in the DNA replication checkpoint, blocking cell-cycle progression through the inactivation of mitotic CDK [105]. Apparently, it is difficult to reconcile this role as an important mediator of the checkpoint response with the mentioned checkpoint-induced Swe1 degradation. One hypothesis is that these two aspects could respond to a different timing and/or signal intensity. It is possible that Swe1 is necessary at the beginning of the response to stop proliferation and only later on, when switching off apical growth is required, is Swe1 degraded. This scenario opens the possibility that the positive regulation of Swe1 by Slt2 in the context of the morphogenesis checkpoint, could be also relevant in the DNA integrity checkpoint. Thus, Slt2, as the DNA integrity checkpoint, could play a positive or negative role on Swe1, to stop proliferation or to switch from apical to isotropic bud growth, respectively. In any case, although defects in the control of Swe1 stability have been related to HU sensitivity [106], *slt2* sensitivity to genotoxic agents was not suppressed by *swe1* deletion [102]. This indicates that Slt2 must affect cell survival to genotoxic stress through an Swe1-independent mechanism.

In this sense, our group has observed that *slt2* hypersensitivity to genotoxic stress is, in fact, suppressed by Cyclin C (also known as Cnc1/Ssn8/Ume3/Srb11) deletion (unpublished results). This result mimics what occurs under oxidative stress: cyclin C deletion suppresses the hypersensitivity of *slt2* mutants to oxidative damage [107]. In response to oxidative stress, Slt2 phosphorylates cyclin C, leading to its translocation from the nucleus to the cytoplasm, where it promotes programmed cell death through extensive mitochondrial fragmentation before its degradation [108,109,110]. It is tempting to speculate that this same mechanism could also occur in the response to genotoxic stress. In support of this idea, an autophagy pathway that is specific to the DNA damage response, called genotoxin-induced targeted autophagy (GTA), has been described [111]; furthermore, Slt2 is required for the optimal induction of autophagy in response to DNA damage, although no specific targets have been identified to date [112].

Another target of Slt2 is the checkpoint adaptor Mrc1. Mrc1 is also a basic regulatory component of the replication complex responsible for maintaining the replication fork progression rate [113] and replication initiation [114]. Slt2 is capable of delaying DNA replication through Mrc1 phosphorylation in response to increased transcription under heat-stress conditions [115]. The authors have proposed the existence of a general S-phase control mechanism mediated by Mrc1, that serves to prevent genomic instability when outbursts of transcription or unscheduled transcription occur during the S-phase [115]. Redundancy between checkpoint adaptor proteins has been proposed recently, suggesting that Mrc1 could also mediate the activation of Rad53 outside of the S-phase [116,117]. This opens the possibility that Mrc1 regulation by Slt2 may be important beyond the S-phase.

Finally, another possible candidate to mediate the role of Slt2 in the DNA damage response could be the Start repressor Whi7. Interestingly, the *WHI7/SRL3* gene has been identified as a suppressor of the *rad53* checkpoint mutant [118] and Whi7 has been localized in gene promoters after HU treatment [119]. This indicates that Whi7 is somehow part of the cellular response to genotoxic stress. As previously mentioned, expression of the *WHI7* gene is almost totally dependent on Slt2 [36,40]. It is reasonable to propose that the regulation of *WHI7* expression by Slt2 could have relevance in the DNA damage response, although this aspect has not yet been tested.

### 7.2. Pkc1 and the DNA Damage Response

The first CWI pathway component that was related to DNA metabolism was Pkc1 [120]. In particular, a *pkc1* mutant strain showed an elevated rate of mitotic recombination. The recombination increase was not rescued by osmotic stabilizing agents, suggesting that Pkc1 regulates DNA metabolism by an alternative pathway not related to the cell wall. This hyper-recombination phenotype was not observed in an *slt2* mutant, which led to the suggestion that this is a specific function of Pkc1, independent of the MAPK cascade. However, recent works have reported that the inactivation of Slt2 [115] or Bck1 [121] results in a significant increase in foci of Rad52 protein, which is considered a marker for homologue recombination [121]. This suggests that the MAPK cascade is involved in some aspect of recombination. In fact, in the case of the *bck1* mutant, although the appearance of Rad52 foci was not associated with a general hyper-recombination phenotype, an increased recombination rate was indeed observed in rDNA, supporting a region-specific role of Bck1 [121]. This could likely be the case for Slt2 as well, which could explain why the *slt2* mutant was not reported in the first study, which assayed the recombination rate using an *ADE2*-locus sectoring assay.

Another relevant connection of Pkc1 with DNA metabolism is that Pkc1 phosphorylates and activates the CTP synthetase [122,123], an enzyme involved in nucleotide biosynthesis. However, CTP synthetase regulation by Pkc1 phosphorylation may be more complex. CTP synthetase contains four putative Pkc1 phosphorylation consensus sites. The mutation of three of them (Ser^36^, Ser^354^, and Ser^454^) to Ala resulted in reduced enzymatic activity and reduced cellular levels of CTP. On the contrary, mutation of the fourth site (Ser^330^) to Ala resulted in increased enzymatic activity and increased cellular levels of CTP [124]. Consistent with a role of Pkc1 in nucleotide biosynthesis, a proteomic study detected phosphorylation of the Rnr2 and Rnr4 subunits of ribonucleotide reductase when a hyperactive Pkc1 is overexpressed [125]. It would be interesting to analyze which residues of CTP synthetase are phosphorylated in vivo by Pkc1 and to determine the effect of this phosphorylation, as well as Rnr2/Rnr4 phosphorylation, and Pkc1 inactivation, on the cellular levels of nucleotide pools. This will help to obtain a better picture of how Pkc1 is implied in nucleotide biosynthesis regulation, an essential issue in the cellular response and survival to genotoxic stress.

On the other hand, the *pkc1* mutant is hypersensitive to different compounds that cause DNA damage, such as methyl metanosulfonate (MMS), 4-nitroquinoline 1-oxide (4NQO) [126], bleomycin [127], and hydroxyurea (HU) [101]. This clearly supports the role of Pkc1 in the cellular response to genotoxic stress. Further supporting this, Pkc1 is phosphorylated in response to DNA damage in a manner dependent on the checkpoint kinases Tel1/Mec1 [128,129]. Under genotoxic stress conditions, Mec1/Tel1 mediate the interaction between Pkc1 and the casein kinase 1 Hrr25. Inactivation of Hrr25 or mutation of Hrr25 consensus sites in Pkc1 abolish Pkc1 phosphorylation, suggesting that Hrr25 is the bona fide kinase responsible for Pkc1 modification [128]. Cells expressing a mutant Pkc1 protein with all the Hrr25 consensus sites mutated showed the same viability as wild-type cells under HU and UV treatments, questioning the physiological relevance of Pkc1 phosphorylation to survive under stress conditions. However, these cells and cells depleted for Pkc1 manifested a defect in the expression of the *RNR3* gene (coding for a large subunit of ribonucleotide reductase), both under basal and genotoxic stress conditions [128]. This result points to a role of Pkc1 in the DNA damage transcriptional response, although the nature of this connection is unknown at present.

Importantly, Pkc1 has also been related to the activation of the DNA integrity checkpoint. Our group has described that Pkc1 activity is required for the activation of the DNA integrity checkpoint in response to different genotoxic stresses in conditional and deleted *pkc1* mutant strains [129]. This result has been questioned by the group of Dr. Levin, who described checkpoint activation in distinct *pkc1∆* strains [103]. We do not know the exact reason for this apparent discrepancy, but it could be due to the different experimental conditions used. Liu and Levin used harder stress conditions involving longer incubation times or higher doses of the genotoxic agents than normally used. We have reproduced their conditions with a *pkc1∆* strain and observed that, although checkpoint activation could be detected, it was severely impaired compared to the wild-type strain. In fact, accurate inspection of the Western blots in [103] showed a mild (but detectable) effect of Pkc1 deletion on checkpoint activation when compared to wild-type strains. A way to reconciliate both results is that Pkc1 was not strictly necessary, but may be required for optimal activation of the checkpoint. Further work is needed to definitively clarify this point. 

A fact that supports DNA integrity checkpoint control by PKC is that this mechanism is not restricted to yeast, but must be a general trait of eukaryotic cells. Humanized yeast expressing PKCδ revealed that this mammalian isoform is able to rescue the checkpoint defect of a *pkc1* mutant strain [129]. Moreover, PKCδ activity is relevant for checkpoint activation by DNA damage in human HeLa cells [129], as well as in mouse NSC (neural stem cells) and ES (embryonic stem) cells, as deduced from the diminished activation of checkpoint effector kinases in the absence of PKCδ (our unpublished results). Other observations support a link between PKCδ and the DNA damage checkpoint. For instance, in the presence of DNA damage, the ATM checkpoint kinase functions upstream of PKCδ activation, inducing PKCδ translocation into the nucleus where it targets the hRad9 protein, a component of the 9-1-1 checkpoint clamp complex, in order to activate apoptosis [130]. PKCδ has also been situated upstream of nuclear DNA-PK and ATM [131,132], and blocking its activity inhibited the phosphorylation of ATM and histone H2AX [131,133]. Moreover, overexpression of PKCδ induced S-phase arrest and activation of the DNA integrity checkpoint [134]. Finally, a recent functional proteomic analysis has proposed additional targets of PKCδ involved in the DNA damage response, DNA repair, and cell-cycle checkpoint activation [135]. The role of PKCδ in regulating cell survival or death has been under discussion for years, but an increasing collection of studies is now illustrating its ability to induce cell-cycle arrest in response to DNA damage at the G1 phase, through the induction of p53 and p21 [136,137,138], S-phase [134], or G2/M phase [139]. All of these observations reinforce the idea that PKC acts in the DNA integrity checkpoint response, in order to brake cell-cycle progression.

In conclusion, the CWI pathway and, more specifically, Pkc1 and the MAPK Slt2 are new players in the cellular response to DNA damage, although their roles are far from being fully understood. In this study, we collected evidence and insights on the possible functions of both proteins, as summarized in Figure 2. Future work will clarify the molecular basis and the relevance for cell-cycle regulation and cell survival of Pkc1, as well as the action of Slt2 in response to DNA damage.

## 8. Concluding Remarks

Cells use signaling pathways to detect and adapt to adverse conditions. The CWI pathway is activated in response to a wide variety of stimuli coming from the outermost structure (the cell wall) to the heart of the nucleus (the genetic material). In response to signals as diverse as plasma membrane/cell-wall damage caused by different compounds, defects in enzymatic activities involved in the biosynthesis of cell-wall components, defects in the actin cytoskeleton or membrane trafficking at the sites of polarized growth, growth in poor medium and low osmolarity, nutrient deficiencies, high temperature, or damage to the DNA and replicative stress, the CWI pathway is able to act on the cell cycle as a canonical checkpoint regulator: it arrests cell-cycle progression until the damage is repaired and the cell can resume cell division with all the guarantees for cell survival. In this scenario, the CWI pathway has been revealed as a versatile toolbox for the cell, with different mechanisms acting on a diverse group of cell-cycle regulators at specific cell-cycle transitions.

## Figures and Tables

**Figure 1 jof-07-01041-f001:**
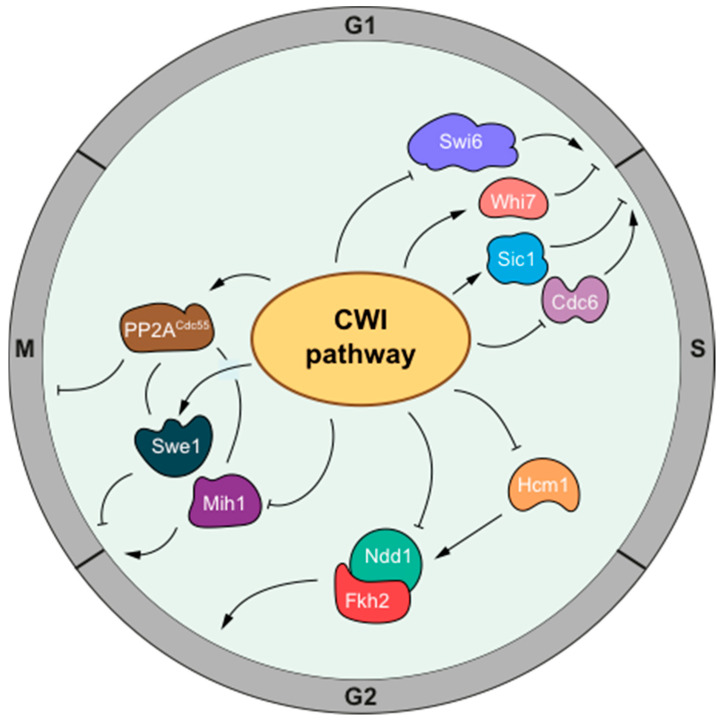
Cell-cycle regulation by the CWI pathway in response to perturbations in the cell surface. This scheme represents the mechanism by which the CWI pathway impinges on cell-cycle regulators to respond to stresses in the cell surface. The CWI pathway has a negative role in the Start transcriptional program, through inducing the expression of the transcriptional repressor Whi7 and promoting the nuclear export of the transcription factor Swi6. It also negatively regulates the initiation of DNA replication, by stabilizing the CKI Sic1 and inhibiting the DNA replication factor Cdc6. The CWI pathway plays a negative role in the G2 transcriptional program, inhibiting the Hcm1 and Ndd1 transcription factors, which results in the impaired expression of mitotic cyclins, among many other mitotic genes. Finally, it affects other aspects of mitotic entry and progression, acting by regulating Swe1, and probably Mih1, to control the inhibition of CDK Cdc28 through Tyr-19 phosphorylation, and by the activation of phosphatase PP2A^Cdc55^, which positively and negatively affects Swe1 and Mih1 and inhibits mitotic progression. Although some molecular details and the specific relevance to cell-cycle regulation of some of these mechanisms are yet to be fully characterized, a scenario has emerged in which the CWI pathway is an important player mediating the arrest of the cell cycle in response to cell-wall/plasma membrane stresses.

**Figure 2 jof-07-01041-f002:**
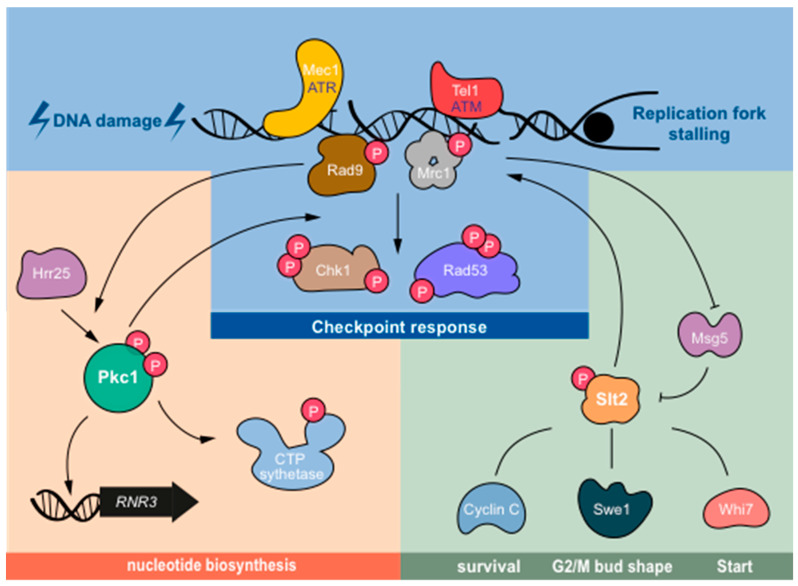
Pkc1 and Slt2 connections with the DNA damage response. This scheme represents the possible roles of Pkc1 and Slt2 related to the response to genotoxic stress. Pkc1 participates in the optimal activation of the DNA integrity checkpoint upstream or downstream of the Mec1/Tel1 apical kinases. Mec1/Tel1, in turn, mediate phosphorylation of Pkc1 by Hrr25 kinase after DNA damage. This regulation is involved in the expression of the *RNR3* gene. Pkc1 is also linked to nucleotide biosynthesis, by regulating the CTP synthetase. Slt2 is activated by genotoxic stress through the inactivation of phosphatase Msg5. Possible Slt2 effectors include: Swe1, which is important for bud morphogenesis and G2/M arrest, whose DNA-damage induced degradation depends on Slt2; Cyclin C, an inducer of programmed cell death responsible for the hypersensitivity of *slt2* mutants to genotoxic treatments; Whi7, a Start repressor regulated by Slt2 whose overexpression suppresses the checkpoint mutation; and Mrc1, a DNA integrity checkpoint adaptor protein that is regulated by Slt2 to stop DNA replication under certain conditions. Although the details of most of these connections are unknown, a scenario is beginning to emerge in which Pkc1 and Slt2 are important players in the response to genotoxic stress.

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
