# Peer review of "The CWI Pathway: A Versatile Toolbox to Arrest Cell-Cycle Progression"

_jof, 2021, doi:10.3390/jof7121041_

Round 1
Reviewer 1 Report
This is a beautiful review of the various roles that the CWI pathway plays in the regulation of the cell cycle. Because this is a difficult topic, characterized by subtle, often secondary impacts, it has been challenging to bring together a coherent understanding of the ways in which this pathway links growth and morphogenesis to the cell cycle. These authors have done an admirable job to tackling what is known and assimilating it into a few manageable areas to regulation. The two figures, one dedicated to the CWI pathway impact on different stages of the cell cycle, and the other to the roles of CWI components in the response to DNA damage, are very nicely set out and easy to understand.
The manuscript requires light editing for English, but is otherwise very good.
Author Response
We thank the Reviewer for his/her comments.
Now, the manuscript has been revised by the English editing services from MDPI.
Reviewer 2 Report
The review at hand summarizes the data on possible relations of CWI signaling in yeast with regulation of the cell cycle at different checkpoints. As such, the manuscript is interesting and well organized. With regard to the content, I have very little to add. However, I would strongly suggest that the authors submit the manuscript to a professional editing service, as there is a considerable amount of “Spanglish”. Thus, singular and plural are frequently mixed up and sentences are often constructed in a complicated manner, which make the manuscript hard to read in parts. Only a few of many examples:
line 447 could be simplified by stating “defects or alterations of the actin cytoskeleton, Swe1 degradation is deregulated and the protein accumulates ....”
line 458: “... of mitosis, Pkc1 phosphorylates the endosulphin ...”
line 470: “... entry into mitosis, inhibiting Cdc28 through phosphorylation at Tyr19 ...”
Throughout the manuscript, proteins are phosphorylated at, not in, specific amino acid residues.
Language should also be checked carefully with regard to the term “expression”. Only genes are expressed, whereas proteins are produced.
Line 239: “of Rlm1 in repairing such damage” Rlm1 is a transcription factor which cannot repair anything. The authors may want to state the Rlm1-mediated transcription is induced in order to compensate for the damage caused.
Lines 622ff: This statement is not clear. The same phosphorylation of Swe1 by Slt2 is proposed to lead to stabilization in one condition and degradation in another? – Isn’t it more likely that there is a more indirect relationship? – Anyway, this statement needs to be rephrased.
Line 630: The official name of the cyclin is Ssn8 according to the SGD database. In the context following and the references cited by the authors , it is also referred to as Cnc1. Srb11 is only one of many other synonyms and should either be omitted or mentioned with all others.
The list of references has apparently not been looked at by the authors before submission of the manuscript. It is full of mistakes both in the titles of the cited references (writing of small and capital letters, italics etc.) and the abbreviations of the journals.
Suggestions for the content:
- It would be very helpfull to have some more detailed figures in the description of complex relations, e.g. one for the paragraph starting in line 262, another for the one starting in line 353, another in line 411. The two figures appearing near the end of the manuscript are nice, but Fig. 1 seems over-simplified (but could be maintained, if the detailed figures suggested are inserted above). The current Fig. 2 is only marginally mentioned in the text. The authors may want to move it up, where the details in the figure are discussed in the text.
- The statement that the function of the CWI sensors is still unclear should be rephrased. The sensors have long been proposed to work as mechanosensors upon clustering in membrane microdomains (Kock et al. 2016, Cell Microbiol. 18, 1251) and for the homologue of Wsc1 in S. pombe the mechanosensor function has been elegantly demonstrated in a recent work (Neeli-Venkarta et al. 2021, Dev. Cell 56, 2856). The authors may want to cite these works.
Author Response
Now, the manuscript has been revised by the English editing services from MDPI.
We appreciate the reviewer for pointing out the problem with the references list. We used Endnote but it is clear that the import filters we used were not appropriate. Now, we have carefully checked the bibliography list.
Cyclin C includes now other name alias most frequently used: ‘Cyclin C (also known as Cnc1/Ssn8/Ume3/Srb11)’
Suggestions:
1) We thought carefully about the design of the figure 1. The main idea we wanted to reflect is that the CWI pathway can affect multiple cell cycle regulators along the cycle. We believe that to introduce more specific details for each mechanism would make the figure too complex and distract from this main message. The same applies to the possibility of introducing single pictures for each mechanism. This is the reason why we prefer to keep the figure in its current state.
We intentionally mention the figures at the end of each block. We want to highlight the figures as a summary/conclusion of the mechanisms previously commented in the text. Thus, we prefer that the first view of Figure 2 reflects this summary character, this is the reason why it is cited at the end of the section.
2) In the original sentence stating that the function of CWI sensors is still unclear, we specifically referred to their role in the morphogenesis checkpoint. In any case, we have removed the sentence because it was unnecessary, the point here is just to indicate that neither Wsc1-3 nor Mid2 are involved in the morphogenesis checkpoint.